# From the Outside to the Inside: New Insights on the Main Factors That Guide Seed Dormancy and Germination

**DOI:** 10.3390/genes12010052

**Published:** 2020-12-31

**Authors:** Chiara Longo, Soyanni Holness, Veronica De Angelis, Andrea Lepri, Sara Occhigrossi, Veronica Ruta, Paola Vittorioso

**Affiliations:** Department Biology and Biotechnology C. Darwin, Sapienza Università di Roma, P.le Aldo Moro 5, 00185 Rome, Italy; longo.493622@studenti.uniroma1.it (C.L.); soyanni.holness@uniroma1.it (S.H.); deavero96@gmail.com (V.D.A.); lepri.1689102@studenti.uniroma1.it (A.L.); sara.occhigrossi1996@gmail.com (S.O.); veronica.ruta@outlook.it (V.R.)

**Keywords:** seed dormancy, germination, light-mediated process, abiotic stress, epigenetic control, translational control

## Abstract

The transition from a dormant to a germinating seed represents a crucial developmental switch in the life cycle of a plant. Subsequent transition from a germinating seed to an autotrophic organism also requires a robust and multi-layered control. Seed germination and seedling growth are multistep processes, involving both internal and external signals, which lead to a fine-tuning control network. In recent years, numerous studies have contributed to elucidate the molecular mechanisms underlying these processes: from light signaling and light-hormone crosstalk to the effects of abiotic stresses, from epigenetic regulation to translational control. However, there are still many open questions and molecular elements to be identified. This review will focus on the different aspects of the molecular control of seed dormancy and germination, pointing out new molecular elements and how these integrate in the signaling pathways already known.

## 1. Introduction

The seed is the structure in which a fully-developed plant embryo is dispersed in the environment, and which enables the embryo to survive the period between seed maturation and seedling establishment, to ensure the development of the next generation. Seed appearance during plant evolution is the most complex and successful method of sexual reproduction in vascular plants. The earliest seed plant emerged in the Devonian, 370 million years ago, from a paraphyletic group termed progymnosperms. Seeds are essential to colonize the surrounding environment, and they represent one of the main food sources for humans. 

Seed dormancy, defined as the inability of seeds to undergo germination under optimal conditions, played a crucial role in the evolution of flowering plants. Indeed, dormancy prevents early germination and vivipary, thus enabling seeds’ dispersal in the environment. Dormancy is established during seed maturation and is finely regulated by a plethora of transcription factors interacting in a complex molecular network which in turn controls hormonal levels and signaling. Abscisic acid (ABA) and gibberellic acid (GA) are the phytohormones mainly involved in the induction, maintenance and release of seed dormancy. These hormones act in an antagonistic manner: ABA promotes the establishment of dormancy and is required for dormancy maintenance while GA triggers dormancy release. Seed germination will then take place properly as for place and time. Indeed, this process only occurs when a special combination of environmental optimal conditions such as light, temperature and water availability are present [1]. Seed germination, in *Arabidopsis* and most plant species, needs a pulse of red light to activate the photoreceptor, which for this process is mainly represented by phytochrome B (phyB) [2]. Active phytochromes promote seed germination also through the control of ABA and GA levels [3,4,5]; indeed, light induces GA biosynthesis and ABA catabolism while repressing GA catabolism and ABA biosynthesis, resulting in increased GA levels and reduced ABA levels. Therefore, the ABA/GA ratio, rather than ABA and GA levels, establishes whether the seed germinates or remains quiescent.

The molecular mechanisms underlying these developmental processes have been deeply analyzed by genetic, molecular and proteomic studies, nevertheless the complexity of these regulatory mechanisms is such that many issues still remain to be elucidated. Several evidences, including recent ones, have highlighted a key role of the epigenetic mechanisms in the negative control of embryo maturation programs in seedlings [6,7,8,9]. While recent studies have highlighted the interaction between environmental cues and chromatin remodeling machinery [10,11], on the other hand, much attention has been paid to translational regulatory mechanisms which play a crucial role during the transition from seed dormancy to seed germination [12].

This review aims to highlight the new regulatory elements of this crucial developmental phase transition, with a particular attention on the crosstalk between the environment and the molecular mechanisms controlling these developmental steps.

## 2. Light Control of Seed Dormancy and Germination

Seed germination is influenced by various environmental cues, the main being temperature, water and light. In particular, red light is an essential requirement for germination of seeds of *Arabidopsis* and most annuals. Among the phytochromes, phyB plays a key role in the promotion of seed germination [13].

In *Arabidopsis thaliana*, PHYTOCHROME INTERACTING FACTOR 1 (PIF1/PIL5), a member of the bHLH family, negatively regulates light-mediated seed germination; indeed, inactivation of *PIF1* partially compensates the inability to germinate of *phyb* mutant seeds, leading to seed germination under red light [14]. The interaction of PIF1 with the active form of phyB induce PIF1 phosphorylation, thus triggering its degradation via the ubiquitin-proteasome pathway, and in turn promoting seed germination [14,15,16]. PIF1 induces transcription of the DELLA encoding genes *GA-INSENSITIVE* (*GAI*) and *REPRESSOR OF GA1-3* (*RGA*) in seeds kept in the dark [17]; GAI and RGA negatively control GA biosynthesis while promoting ABA biosynthesis under dark conditions [17]. GAI cooperates with the DOF transcription factor DOF AFFECTING GERMINATION 1 (DAG1), which represses seed germination through directly binding the promoter of the GA biosynthetic gene *GIBBERELLIN 3-OXIDASE 1* (*GA3ox1*), to inhibit its expression [18]. Similarly, the MYB transcription factor REVEILLE 1 (RVE1) represses the expression of the GA biosynthetic gene *GA3ox2*, to decrease the level of bioactive GAs [19]. Interestingly, it has been recently demonstrated that RVE1 interacts, both in vitro and in vivo, with RGA LIKE 2 (RGL2) [20], the DELLA protein primarily involved in the repression of GA-mediated seed germination [21]. RVE1 also interacts with the E3 ubiquitin ligase SLEEPY1 (SLY1), which mediates RGL2 degradation in the presence of GA, to suppress RGL2-SLY interaction and consequently increasing RGL2 stability [20]. Transcriptomic analysis has shown that RVE1 and RGL2 synergically regulate gene expression of several targets including a set of genes involved in GA biosynthesis as well as ABA, auxin and ethylene signaling pathways [20]. Although it was suggested that the phyB-RVE1-RGL2 pathway was independent of the PIF1 pathway [20], recent studies proved that PIF1 interacts with RVE1 [22]. Moreover, genetic analysis revealed that RVE1 and PIF1 additively control phyB-mediated seed germination, since *pif1rve1* double mutant seeds showed higher germination rates than the single mutants; RVE1 repression of this process requires PIF1, whereas PIF1 activity is at least partially dependent on RVE1. In addition, *RVE1* and *PIF1* expression levels are mutually and positively regulated [22], thus outlining a complex signaling pathway with PIF1 and RVE1 as upstream master regulators which control target genes as *GA3ox1* and *GA3ox2,* through the activity of the DELLA proteins RGL2 and GAI (Figure 1).

The VASCULAR PLANT ONE-ZINC FINGER 1 and 2 (VOZ1 and VOZ2) were recently added as downstream elements of the phyB-PIF1 signaling pathway; VOZ1 and VOZ2 transcription factors have been previously characterized as phyB-interacting proteins which redundantly promote flowering [23]. Interestingly, it has been demonstrated that *voz1voz2* double mutant seeds—conversely to the single *voz1* and *voz2* mutants—show a higher germination rate than the wild type under phyB-off conditions, (after a pulse of far red light), suggesting that the VOZ proteins redundantly repress phyB-mediated seed germination [24]. The expression level of both *VOZ* genes is downregulated during imbibition in the light; in addition, in *pif* mutant seeds, expression of *VOZ* genes was reduced irrespective of light conditions, indicating that VOZ1 and VOZ2 act downstream PIF1 which induces their expression in the dark [24]. By means of a ChIP assay, it was proved that VOZ2 directly binds the *GA3ox1* promoter to repress its expression, consistent with increased expression level of the *GA3ox1* gene in *voz1voz2* double mutant seeds [24]. Interestingly, this scenario is very reminiscent of the one previously shown for the DAG1 protein, which is induced indirectly by PIF1 and, in turn, represses *GA3ox1* expression to inhibit phyB-dependent seed germination [18,25]. It is tempting to figure a signaling pathway underlying light-mediated seed germination in which all these molecular elements might merge (Figure 1); further studies will be needed to unveil the genetic and molecular relationships between PIF1 and downstream factors, as well as between DAG1 and VOZs, or RGL2 and GAI. The result will be of remarkable interest not only for basic research but also to improve the yield of species of agronomic interest.

## 3. Hormonal Control of Seed Dormancy and Germination

Dormancy and germination of seeds are two processes finely regulated by several phytohormones; indeed, although ABA and GA play the main role, auxin, cytokinins (CKs), and jasmonate (JA) have been shown to partly contribute to seed germination [26,27,28,29]. As for brassinosteroids (BRs), the involvement of this class of molecules in the promotion of germination has been shown since a long time [30]. Interestingly, it was recently proved that the transcription factor BRI1-EMS-SUPPRESSOR1 (BES1), which is part of the BR signaling pathway, physically interact with the ABA-responsive bZip transcription factor ABA INSENSITIVE5 (ABI5) [31], to restrain ABI5 from binding the promoters of target genes, thus promoting seed germination [31].

Additionally, the gaseous hormone ethylene plays a role in the control of both dormancy and germination of seeds [32,33,34]. Previous studies have shown that ethylene stimulates dormancy release and seed germination in dicot species, while inhibition of ethylene synthesis is related with repression of germination [32,33]. Consistently, inactivation of the membrane-associated receptor *ETHYLENE RESPONSE1* (*ETR1*) and the downstream factor *ETHYLENE INSENSITIVE 2* (*EIN2*) results in more dormant mutant seeds compared to wild-type seeds [35,36,37]. It has been recently demonstrated that the *reduced dormancy 3* (*rdo3*) loss-of-function mutant [38] is an *etr1* mutant allele [39]. *rdo3* was isolated for its reduced dormancy; further analysis revealed that *rdo3* mutant seeds were not altered in ABA sensitivity or endogenous ABA levels [40]. The recent study by Li et al. [39] proved that ETR1 promotes the establishment of seed dormancy in Arabidopsis, and its function requires DELAY OF GERMINATION1 (DOG1), which has been previously identified as a major quantitative trait locus controlling seed dormancy [41]. The activity of DOG1 in the promotion of seed dormancy is strictly dependent on ABA signaling; indeed, DOG1 controls dormancy at least in part through the control of *ABI5* expression [42,43]. Analysis of transcriptomic data of the *rdo* mutant led to identify ETHYLENE RESPONSE FACTOR12 (ERF12) as a downstream element; indeed, lack of ETR1 results in an increased *ERF12* transcript level, suggesting that ERF12 is involved in the ETR1-mediated dormancy, and it is likely to represent a link between ETR1-ethylene and the DOG1 pathway in the regulation of seed dormancy in *Arabidopsis*. ERF12 belongs to the ERF subfamily of repressors [44], which interact with the TOPLESS (TPL)/TPL-related (TPR) corepressors [45,46,47]. TPL does not bind directly DNA, but is required for DOG1 repression mediated by ERF12, as demonstrated by luciferase assay [39]. Although the molecular elements between the ETR1 receptor and ERF12 are still unidentified, these findings uncovered, at least in part, the molecular pathway which controls seed dormancy, linking ethylene to ABA signaling through ETR1-ERF12/TPL and DOG1. Interestingly, ETR1 is likely to be involved also in the repression of seed germination; indeed, a previous study revealed that *etr1* mutant seeds exposed to far-red light or in darkness, showed increased germination rate compared to wild type seeds [48]. Surprisingly, this germination behavior was not dependent on altered endogenous ethylene levels between mutant and wild-type seeds, but on increased GA and reduced ABA levels in *etr1* mutant seeds following far red light treatment [48].

Seed germination and seedling establishment have been frequently considered as two distinct developmental processes, despite they share both hormonal cues and molecular elements. Among phytohormones, ABA is known to inhibit both seed germination as well as post-germination seedling development; ABA repression requires the ABA-responsive transcription factor ABI5 [49,50]. It has recently been shown that the ABA-ABI5 repression of the seed-to-seedling transition is inhibited by the bZIP transcription factor ELONGATED HYPOCOTYL5 (HY5) [51], the main positive regulator of photomorphogenesis [52,53]. Based on genetic and phenotypic analysis of different *hy5* mutant alleles, Yadukrishnan et al. [51] showed that *hy5* mutants display an ABA hypersensitive phenotype both for seed germination and seedling development. In addition, it has also been established that HY5 acts downstream the E3 ubiquitin ligase CONSTITUTIVELY PHOTOMORPHOGENIC1 (COP1) to repress the ABA-mediated inhibition of seedling development [51]. COP1 mediates HY5 ubiquitination and subsequent proteasomal degradation in darkness [54]; COP1 belongs to the superfamily of the WD-repeat (WDR) proteins, as it is characterized by a WD-40 protein-protein interaction domain [55]. Interestingly, it has recently been identified a WD-40 ABA-induced protein, ABA Signaling Terminator (ABT), which has been proved to counteract ABA inhibition of seed germination and seedling establishment [56]. By analysis of ABA sensitivity of *abt* mutant alleles (one T-DNA and one Crispr-Cas9, respectively), of the *abt-D* dominant allele and of the *ABT* overexpressing line *ABT-OE* during germination and seedling development, it has been proved that ABT function as a negative regulator of ABA activity during the seed-to-seedling transition [56]. ABT physically interacts with the PYR/PYL ABA receptors and with the downstream signaling molecules ABA INSENSITIVE 1 and 2 (ABI1 and ABI2); the interaction of ABT with ABI1/2 interferes with the binding of PYR/PYL with ABI1/2, thus impinging on the ABA signaling pathway. ABT is also induced by ABA, suggesting a feedback negative loop, where ABT is produced to release ABI1/2 inhibition which leads to induction of ABA-responsive genes through activation of the SNF1 (SUCROSE NON-FERMENTING 1)-related protein kinase 2 (SnRK2). This regulatory mechanism should counteract overactivation of ABA responses [56]. Similarly to the dormancy-germination transition, GA has an opposite function to ABA also in the seed-to-seedling transition; indeed, it has been recently shown that GA is required for seedling development [57], as well as for radicle protrusion from the seed coat, as previously reported [58]. In this recent study it was demonstrated that the inability to germinate of *ga1-3* mutant seeds can be rescued to a certain extent by overexpression of the GA receptor GIBBERELLIN INSENSITIVE DWARF1 (GID1), but it leads to aberrant seedling development, unless supplied with exogenous GA [57].

A recent study showed that reduced ethylene levels coupled with decreased starch degradation in the endosperm are linked to a reduction of embryo axis, coleoptile, and root growth; this suggests that ethylene also regulates seedling growth in wheat partly through transcriptional control of storage starch degradation [34]. It was also demonstrated that ethylene acts through spatial and temporal modulation of the ABA/GA balance [34]. These effects were shown to be a consequence of the control of GA levels and sensitivity, through expression of GA metabolic (*TaGA20ox1*, *TaGA3ox2*, and *TaGA2ox6*) as well as signaling (*TaGAMYB*) genes; indeed, endospermic bioactive GA promotes starch degradation through transcriptional regulation of *TaAMY1*, *TaAMY3*, and *TaAMY4*, specific α-amylase genes, and *TaAGL1*, an α-glucosidase gene, while GA in non-endospermic tissues induces cell wall expansion through expression of the specific α-expansin genes *TaEXPA3*, and/or *TaEXPA7*, and/or *TaEXPA9* [34]. Furthermore, ethylene represses both ABA level and signaling in non-endospermic tissues through the expression of ABA metabolic (*TaNCED2, TaCYP707A1*), and signaling (*TaABI3, TaABI5*) genes, thus contributing to the induction of cell wall expansion through expression of *TaEXPA* genes [34].

## 4. Abiotic Stress Effects on Seed Germination

Throughout the whole life cycle, plants have to face changing environmental conditions; this is particularly important for the seed germination process, which, if occurring under unfavorable environmental conditions, can compromise the propagation of the species and crop yield. Abiotic stresses (such as drought, salinity, and temperature) cause osmotic and oxidative stress, triggering turgor loss, reduced protein activity, and excess levels of reactive oxygen species (ROS). Under these conditions the plant cellular processes, including metabolism, gene expression and enzyme activity, are modified [59,60]. In addition, these events mainly affect the metabolic and hormonal signaling pathways [61,62]. The cellular disorder resulting from abiotic stress can severely affect plant growth and development, from seed germination to flowering.

Cold conditions (i.e., chilling between 0 and 8 °C and freezing below 0 °C) are among the most relevant factors limiting the growth and distribution of plant species [59,63]. Plant response to cold stress is mediated by the INDUCER OF CBF EXPRESSION1 (ICE1)-C-Repeat Binding Factor/DRE Binding Factors (CBF/DREBs)-COLD RESPONSIVE (COR) signaling pathway. The activation of CBFs is not only temperature-dependent, since it is also affected by environmental cues such as light quality and daylight length, and multiple hormonal pathways [64].

Sheepgrass (*Leymus chinensis*) is a forage perennial grass highly tolerant to drought, salt and cold stress; a transcriptomic analysis allowed to identify several stress-induced genes which could provide the genetic basis of the environmental adaptation of this forage grass [65]. Among these stress-related factors LcSAIN1, LcSAIN2, LcFIN1 and LcFIN2 were shown to induce salt and/or cold stress tolerance in transgenic Arabidopsis and rice [66,67]. Recently, the MYB transcription factor LcMYB4 was identified from sheepgrass and shown to be involved in cold stress response. In particular, overexpression of *LcMYB4* in Arabidopsis showed increased cold tolerance during seed germination and seedling development, as revealed by higher germination rate and root length in cold-treated transgenic overexpressing lines compared to cold-treated wild type [68].

The *Arabidopsis* plasma membrane-associated cation-binding protein 2 (PCaP2), involved in the control of directional cell growth and cortical microtubule organization [69], was recently reported to play an important role in chilling tolerance by triggering the CBF regulatory pathway and ABA response through SnRK2-mediated transcriptional pathway [70]. Indeed, *PCaP2*-overexpressing plants displayed enhanced chilling tolerance, whereas *pcap2* RNAi and mutant lines were more sensitive to chilling stress during seed germination, seedling development and reproductive growth [70]. Interestingly, *PCaP2*-overexpressing plants also showed increased tolerance to water stress, whilst *PCaP2* RNAi and *pcap2* mutant lines were less tolerant to water deficit for seed germination, seedling development, and plant survival [71]. Consistently, *PCaP2* gene expression was induced by cold, heat, drought and salt stresses, as well as by ABA and Salicylic acid (SA), thus suggesting that PCaP2, besides its role in Ca binding and microtubule organization, plays a role in the signaling pathways underlying hormone-mediated plant response to abiotic stress. Interestingly, inactivation of *PCaP2* resulted in decreased expression level of the *SnRK2* genes, encoding ABA positive signaling factors, as well as of the SnRK2-mediated ABA-responsive *ABRE-binding factors* (*ABF*) encoding genes, substantiating a positive role for PCaP2 in ABA-mediated response to cold stress in seeds [70].

Just as cold, high temperature (heat stress) can also affect plants growth and development. *Ricinus communis* L. is able to germinate at high temperatures, although further seedling development is negatively influenced. Interestingly, the malate synthase encoding gene (*RcMLS*) plays a major role in seeds’ ability to germinate at high temperatures, consistent with its crucial role in lipid mobilization. *Arabidopsis* is very sensitive to high temperature, wild type (Col) seeds showed 37% germination rate at 35 °C. However, *Arabidopsis RcMLS* overexpressing seeds were able to reach 80% germination under heat stress [72], thus corroborating the essential role of lipid composition in the response to temperature stress during seed germination.

Salinity or salt stress is well known to affect overall plant development. The main effects of salt stress on seed germination are the prevention of water uptake and ionic toxicity [73]. Crop yield of alfalfa (*Medicago sativa* L.), an important forage, is strongly affected by salinity. The ionic/oxidative stress results in a decreased seed germination potential, while the osmotic stress delays seedling growth. Strikingly, ethylene can induce salt tolerance in Medicago seeds, thus increasing germination level under salt stress. This ethylene-mediated salt tolerance is dependent on the ETHYLENE RECEPTOR2 (MsETR2), as silencing of the *MsETR2* receptor gene leads to the loss of the ethylene-triggered salt tolerance [74]. In addition, it was shown that ethylene treatment caused a repression of the ethylene biosynthetic ACC oxidase gene (*MsACO*), possibly via a feedback loop, and of the *Ethylene Response Factor* (*MsERF8*) gene, thus boosting seed tolerance to salt stress. On the other hand, expression of the *MsERF11* gene was slightly but significantly upregulated by ethylene treatment, suggesting a potential positive role in the control of seed germination under salt stress [74].

LRR-only proteins are a sub-family of the LRR protein family, which are characterized by the presence of the single LRR domain [75]. So far, their function is largely unknown; it was recently demonstrated the involvement of *Arabidopsis* LRRop-1 in the response to ABA-mediated abiotic stress response in seed germination. Analysis of *lrrop-1* mutant seeds revealed an increased germination rate in the presence of ABA or NaCl, or of the GA inhibitor paclobutrazol (PAC), suggesting that inactivation of *LRRop-1* results in higher abiotic stress tolerance. Consistently, *lrrop-1* dry seeds showed a reduced amount of endogenous ABA while imbibed seeds displayed an increased GA7 level compared to wild type seeds [76].In addition, in ABA-treated mutant seeds the expression level of the ABA-responsive genes *ABI3*, *ABI5*, and *RESPONSIVE TO DESICCATION 29A* and *B* (*RD29A*, *RD29B*) was significantly decreased compared to ABA-treated wild-type seeds. Interestingly, the DOF6 transcription factor, an ABA-related repressor of seed germination [77], positively regulates *LRRop-1* expression in seeds, corroborating the role of LRRop-1 as an ABA-related repressor of seed germination under abiotic stress.

In addition to stress tolerance, defined as the potential to acclimate to stressful conditions, plants can adopt a stress avoidance strategy, which involves a variety of protective mechanisms to delay or prevent the negative impact of stress factors. An example is the Arabidopsis karrikin receptor KARRIKIN INSENSITIVE2 (KAI2), which has been shown to be involved in the response to osmotic stress, salinity, and high temperature in seeds [78]. Indeed, *kai2* mutant seeds showed a lower germination rate compared to the wild type in response to osmotic stress rather than salinity, indicating that inactivation of *KAI2* results in a higher sensitivity to abiotic stress as well as to ABA. Interestingly, this karrikin-KAI2 signaling pathway promotes germination of the wild-type under suitable conditions but inhibits seed germination under adverse conditions such as osmotic stress and high temperature. The karrikin-mediated seed germination pathway is light-dependent [79] and is likely to be upstream of ABA and GA, suggesting that KAI2 may function as a sensor of environmental cues to determine if a seed can germinate or should retain dormancy, under the control of the main hormones ABA and GA [78].

SKP1 (S-phase kinase-associated protein1) proteins are key subunits of the SCF (SKP–cullin–F-box protein) E3 ligase complexes which target proteins for ubiquitin-mediated degradation in eukaryotes [80,81]. The role of plant SKP1 proteins it is still to be unveiled, although they have been involved in the response to abiotic stress [82,83]. An Arabidopsis SKP1-like protein13 (ASK13) was recently identified to be differentially expressed during seed development and up-regulated in response to abiotic stress (cold, heat, salt, drought). *ASK13* overexpression and knockdown lines proved that ASK13 plays a positive role in seed germination and seedling growth in response to abiotic stress. Indeed, under stress conditions, *ASK13* overexpression results in increased seed germination rate and improved seedling development, whereas *ask13* knock-down mutant seeds showed delayed germination and a reduced germination rate compared to the wild type [84]. Screening for ASK13 interacting proteins revealed that ASK13 can interact with both F-box and non-F-box proteins, suggesting that it can play a role in other processes besides protein ubiquitination [84]. ASK13 can also probably take part in numerous SCF–E3 complexes to control the abundance of various regulatory proteins through ubiquitination; therefore, it will be interesting to unveil if the role of ASK13 in seed germination under abiotic stress conditions is dependent on the protein regulation proteasome-mediated, or on the interaction with other protein complexes.

## 5. Translational Control of Seed Dormancy and Germination

The seed is an autonomous structure in which a fully developed embryo is spread in the environment, allowing the establishment of an autotrophic organism. In *Arabidopsis*, seed development is divided in two major phases: embryo and endosperm development (or morphogenesis), and maturation [85,86]. Once embryogenesis is completed, seeds enter the maturation phase, dormancy is established, storage compounds and mRNAs are accumulated, and seeds become desiccation tolerant [87,88]. Once dormancy is released and the environmental conditions are permissive, seeds can germinate; this step represents a programmed transition from a quiescent to a metabolically active state. Since in the presence of the transcription inhibitor α-amanitin germination can occur, whereas cycloheximide blocks this process, germination of seeds is not strictly dependent on transcription of newly synthetized mRNAs, whilst it requires de novo protein synthesis [89,90,91,92,93,94,95]. The presence of stored mRNAs in dried seeds was discovered 50 years ago [96,97], and so far, they have been detected in a large number of seed species [98,99,100]; nevertheless, only in the last decade many open questions on the seed-stored mRNAs and on the translational control underlying dormancy release and seed germination have been, at least in part, addressed [12].

Genome-wide analysis showed that Arabidopsis mature dry seeds hold more than 12,000 transcripts, whereas rice dry seeds have about 17,000 different stored mRNAs [98,101]; it is assumed that not all these stored mRNAs are required for seed germination and a large number should represent housekeeping genes. Among the transcripts specifically required for seed germination, there are mRNAs related to the translation machinery, as well as ubiquitin and proteasome system, thus corroborating the importance of protein synthesis, and suggesting there should be a dynamic regulation and selective proteolysis during early seed germination [98]. A combined approach based on two-dimensional gel-based differential proteomics and dynamic radiolabeled proteomics demonstrated that germination starts when storage and desiccation tolerance-related proteins are synthesized, to guarantee that germination occurs only under favorable conditions [102]. Interestingly, among the translated mRNAs during the transition phase from seed-to-seedling, there are transcripts from hypoxia stress-related genes, thus pointing out the importance of a molecular control of low-oxygen conditions during germination [103]. ABA and GA control dormancy and germination antagonistically, with the former promoting dormancy and inhibiting germination, and GA inducing release of dormancy and germination; therefore, it is not surprising that, among the most represented stored mRNAs in dry seeds, there are transcripts from ABA-related genes, as they have ABA-regulated motifs or ABA responsive elements (ABREs), suggesting that they are accumulated during the maturation stage [98].

A prominent work on the polysome profiling of Arabidopsis dry to germinating seeds clearly identified two steps where the ratio of polysome-associated respect to total mRNAs shifted, and most significant translational changes occurred: the hydration and germination phases, named HTS and GTS (Hydration Translational Shift and Germination Translational Shift), which can be correlated to the physiological phases of early imbibition and radicle protrusion, respectively [103]. Distinct and not overlapping transcript sets, and corresponding genes, were associated with these translational shifts. A bioinformatic analysis led to the identification of peculiar sequences associated with different translational shifts. The peculiar features were the number of µORFs, transcript length, GC content, and secondary structure which mainly correlate with an upregulation of translation. Interestingly, µORFs in the 5′ UTR was significantly over-represented in the HTS downregulated gene set, while they were significantly under-represented in the GTS downregulated gene set [103], thus suggesting that translational control of the seed-to-seedling transition involves multiple factors and different molecular mechanisms.

Several studies have dealt with the problem of mRNA storage for a long time, but only recently the new methodologies allow to answer this question [104,105]. A recent study established that half of stored mRNAs (50%) in dry seeds are associated with ribonucleic protein complexes, ensuring the survival of mRNAs during seed storage [106]. A part of these transcripts is associated with monosomes and actively transcribed, another portion is included in the polysomes-associated fraction and contains transcripts mainly represented in the GO categories “embryo development ending in seed dormancy”, “pollen development”, “flower development”, and “pollen germination”, thus suggesting that they are not necessary for seed germination. Indeed, a large number of these transcripts are downregulated during the HTS phase. The VARICOSE P-body protein, known to be associated with both monosomes and polysomes, is likely to be involved in the degradation of these transcripts. Consistently, *varicose* mutant seeds show an increased germination rate, indicating that in *varicose* seeds a large number of transcripts required for the beginning of germination are accumulated, whereas they are degraded in wild-type seeds [106,107].

## 6. Epigenetic Control of Seed Dormancy and Germination

The transition from seed dormancy to germination is finely regulated by different partially overlapping mechanisms, among which the epigenetic control plays a key role. Indeed, genes specifying seed development, maturation, and dormancy, as well as genes required for the accumulation of seed storage compounds, must be sequentially repressed by chromatin-based modifications, to allow the embryo to differentiate into an autotrophic organism [108]. Among these genes, the most relevant are *LEAFY COTYLEDON 1* (*LEC1*), *ABA INSENSITIVE 3* (*ABI3*), *FUSCA 3* (*FUS3*), and *LEC2*, termed *LAFL* [8], encoding the master transcriptional regulators of seed development. These factors in turn control a number of maturation and dormancy genes like the storage compound genes, *CRUCIFERIN 3* (*CRU3*), *CRUCIFERINA 1* (*CRA1*), *SEED STORAGE ALBUMIN 1* and *2*, (*SESA1* and *2*), and the dormancy regulators *SOMNUS* (*SOM*), *DELAY OF GERMINATION 1* (*DOG1*) and *DOF AFFECTING FACTOR 1* (*DAG1*) [6,9,109]. The establishment and maintenance of epigenetic repression of all these genes upon germination and vegetative growth is mainly accomplished by PcG (Polycomb Group) proteins through the Polycomb Repressive Complex 1 and 2 (PRC1, PRC2), and by the acetylases/deacetylases system and chromatin remodeling complexes, although the latter are so far less characterized.

PRC2 components and molecular functions are evolutionarily well conserved between plants and animals [110], although in plants the four PRC2 core subunits originally discovered in Drosophila are represented by twelve homologs [111]. These subunits give rise to three complexes, EMBRYONIC FLOWER (EMF), VERNALIZATION (VRN) and FERTILIZATION INDEPENDENT SEED (FIS), with at least partially distinct functions, although they share a certain number of common target genes [6,112]. In contrast, the extent of the evolutionary conservation of PRC1 remained enigmatic for a long time. In Arabidopsis there are six homologs of the *Drosophila* PRC1 subunits, with the lack of a Ph homolog and the protein EMBRYONIC FLOWER 1 (EMF1) as a plant-specific component of PRC1 complexes [113,114].

PRC2 catalyzes the trimethylation of lysine 27 of histone H3 (H3K27me3), which is recognized by PRC1 and establishes stable transcriptional repression by monoubiquitination of histone H2A (H2Aub) [115]. More recently, it has been demonstrated that PRC1 activity can also be required for PRC2 recruitment, and that both complexes can work independently with different functions, depending on the interacting proteins and target genes [116,117,118]. In addition, LIKE HETEROCHROMATIN PROTEIN1 (LHP1), which has been primarily associated with PRC1 function [119], was also found to be associated with members of PRC2 [120], querying the strict division of both complexes in *Arabidopsis* (Figure 2).

Consistent with the requirement of PRC1 and PRC2 activity for the repression of embryo maturation programs in seedlings [6,7], mutations affecting PRC1 or PRC2 lead to a switch from embryo to seedling development caused by lack of *LAFL* genes repression; expression of these genes, in wild type seeds, is induced through the H3K4me3 activation marker during seed development and silenced upon seed germination by the H3K27me3 repression marker [6,121]. However, how seed developmental genes timely switch from activation to a repression chromatin state is not yet fully unveiled. It has been proposed a role for the ALFIN1-like (AL) PHD-domain proteins, which read H3K4me2/me3 marker and trigger the recruitment of PRC1 components via AL-AtBMI1, AL-AtRING1, and AtBMI1-AtRING1 direct interactions, forming an AL PHD-PRC1 complex, that lead to a switch from H3K4me3 to H3K27me3 [121]. The ZUOTIN-RELATED FACTOR1 (ZRF1) proteins are also reader of the H2Aub1 marker, characterized by the SANT (Swi3, Ada2, NcoR1, and TFIIIB) domain present in several chromatin modifiers [122,123]. ZRF1a/b are required for the suppression of the expression of seed developmental genes after germination; indeed, the *atzrf1a atzrf1b* mutants show derepression of seed-specific genes, similarly to the *atbmi1a atbmi1b* PRC1 mutants. AtZRF1a/b-PRC1/PRC2 interactions are likely involved in the spreading of H2Aub1 and H3K27me3 deposition [124].

PRC2 plays a crucial role also in the balance of ABA and GA. High ABA and low GA levels lead to seed maturation allowing the establishment of dormancy. The relationship is reversed during germination by PRC2 inhibition of positive regulators of ABA signaling and negative regulators of GA signaling. The CCCH-type zinc finger protein SOMNUS (SOM) plays a role in downregulating GA and upregulating ABA levels [125]. *SOM* is a seed-specific gene and a PRC2 target; its expression is down-regulated through deposition of H3K27me3, in order to increase GA while reducing ABA levels, thus allowing seed germination [6]. It has been recently established that during germination, the epigenetic factor POWERDRESS (PWR) interacts with ABI3 and simultaneously recruits HISTONE DEACETYLASE 9 (HDA9) activity to reduce histone acetylation level and increase H2A.Z deposition at the *SOM* locus, thus repressing its expression level [9,10]. Interestingly, PWR is involved in high temperature tolerance in seeds. Indeed, *PWR* overexpressing lines displayed increased thermotolerance while *pwr* mutant seeds showed decreased thermotolerance during seed germination, thus suggesting that PWR can improve thermotolerance in seeds [10].

*DOG1*, which encodes the major genetic regulator of seed dormancy in Arabidopsis [126], is a PRC2 target, downstream *LAFL* genes. *DOG1* expression is rapidly induced during seed development and the amount of the DOG1 protein is finely regulated during seed development and germination, both at transcriptional and post-transcriptional levels [127,128]. Recent evidences have further elucidated the epigenetic control of *DOG1* expression. The transcriptional repressors HIGH-LEVEL EXPRESSION OF SUGAR INDUCIBLE2 (HSI2) and HSI2-LIKE1 (HSL1) have been shown to be recruited on the *DOG1* promoter in vivo to repress its expression, consistent with the increased *DOG1* transcript level in *hsi2hsl1* double mutant seeds and reduced germination rate of *hsi2hsl1* freshly harvested seeds [129]. The negative control of *DOG1* expression was proved to be associated with deposition of the H3K27me3 mark, as revealed by qPCR-ChIP assay; accordingly, *hsi2* mutant lines showed a significant decrease of H3K27me3 levels compared to the wild type. In the *hsi2hsl1* double mutant the enrichment of the repressive mark was further reduced, thus suggesting that HSL1 contributes to HSI2 activity [129]. The PRC1 subunit LHP1 and the PRC2 catalytic subunit CURLY LEAF (CLF) have been demonstrated to interact in vivo with both HSI2 and HSL1, thus outlining an epigenetic regulation circuit for *DOG1* repression in seeds [129]. The interaction of HSI2 with the PRC2 subunit MULTIPLE SUPPRESSOR OF IRA1 (MSI1) further substantiated its key role in the negative control of *LAFL* genes, besides *DOG1*. Interestingly, MSI1 has been shown to interact with HISTONE DEACETYLASE19 (HDA19) [130], to decrease the expression of ABA receptor genes and, in turn, of ABA-responsive genes [131]. HSL1 has also been shown to interact in vivo with HDA19 to silence seed-specific genes during seed germination [132], whereas HIS2 interacts with the closest homolog HDA6 [133], thus suggesting that, besides methylation, also acetylation levels are crucial to repress seed-specific genes during seed germination. The interaction of both HSI2 and HSL1 with MEDIATOR13 (MED13), a complex which links transcription factors to the RNA polymerase II machinery, further substantiates the role of these proteins in the seed-to-seedling transition. Indeed, the molecular model proposed by Chhun et al. [133] suggests that MED13 is recruited by the homo/heterodimer HSI2/HSL1 on the promoter of the *LAFL* genes; this complex would also recruit HDA19 and HDA6, interacting with HSL1 and HSI2 respectively, to erase the H3K9/K14 mark and strengthening the repression of *LAFL* genes [133] (Figure 2).

The Su(Var)3-9 homolog SUVH5, a methyltransferase belonging to the SUV(R) group of SET domain proteins, which catalyzes the dimethylation of H3K9 in imbibed seeds, has been recently shown to function as a positive element of the light-mediated seed germination process [134]. Indeed, inactivation of *SUVH5* results in a reduced germination rate under phyB-dependent conditions; consistently, a transcriptomic analysis of *suvh5* mutant seeds revealed deregulated expression of 24.6% light-regulated genes. RNA-seq assay and expression analysis revealed that SUVH5 negatively controls both ABA biosynthetic and signaling genes as well as GA catabolic and *DELLA* signaling genes, in order to decrease the ABA/GA ratio [134]. The positive function of SUVH5 was further established by the presence of *DOG1* among the upregulated genes and by the increased level of the four *DOG1-like* genes (*DOGL2-4*), thus indicating that SUVH5 represses these negative regulators to promote light-mediated seed germination [134].

Repression of *DOG1* is also mediated by the ATP-dependent chromatin-remodeling protein PICKLE (PKL), as revealed by overexpression of *DOG1* in a *pkl* mutant background [11]. Inactivation of *PKL* affects also the expression level of *LAFL* genes, which are derepressed in freshly harvested *pkl* mutant seeds, thus suggesting that PKL is required to repress seed-specific genes. PKL binds chromatin on the *DOG1* locus, as does the PKL-interacting protein LUX AR- RHYTHMO (LUX), a member of the Evening Complex (EC) of the circadian clock [135]; indeed, PKL and LUX promote H3K27me3 deposition via PRC2 activity. The EARLY FLOWERING3 and 4 (ELF3 and ELF4) proteins are components of EC; analysis of *DOG1* expression in *elf3* and *elf4* mutant seeds, as well as the seed germination phenotypes of freshly harvested mutant seeds showed that ELF3, and not ELF4, cooperates with PKL in *DOG1* repression. Consistently, the increased *DOG1* expression and dormancy phenotype of *lux* and *elf3* freshly harvested mutant seeds is correlated with the photoperiod applied during growth of the mother plant, with long day (LD) compared to continuous light (CL) conditions resulting in a stronger phenotype [11]. Although several evidences point out a role of the circadian clock in the control of seed dormancy, so far the molecular mechanism involved was still unknown. This study helps to integrate the clock signaling pathway with the regulation of seed dormancy; further studies are needed to combine external stimuli such as light and temperature as well as other molecular elements involved in the control of the circadian clock.

## 7. Conclusions

Seed germination is the first developmental process in plant life cycle. The transition from seed dormancy to seed germination is crucial for the production of offspring and is an important ecological and commercial trait. Despite this developmental step having been deeply studied, new evidence still contributes to further elucidate the complex regulatory mechanism underlying these processes. To ensure the development of the next generation, the molecular network integrates the response to environmental cues with hormonal signals, involving transcriptional, translational, as well as epigenetic regulatory mechanisms.

Although in recent years several studies have contributed to shedding light on the molecular networks underlying these processes mainly in model plants, the use of cutting-edge technologies will allow to deepen the knowledge of these processes in plants living in different habitats, with potential implications of agronomic interest.

## Figures and Tables

**Figure 1 genes-12-00052-f001:**
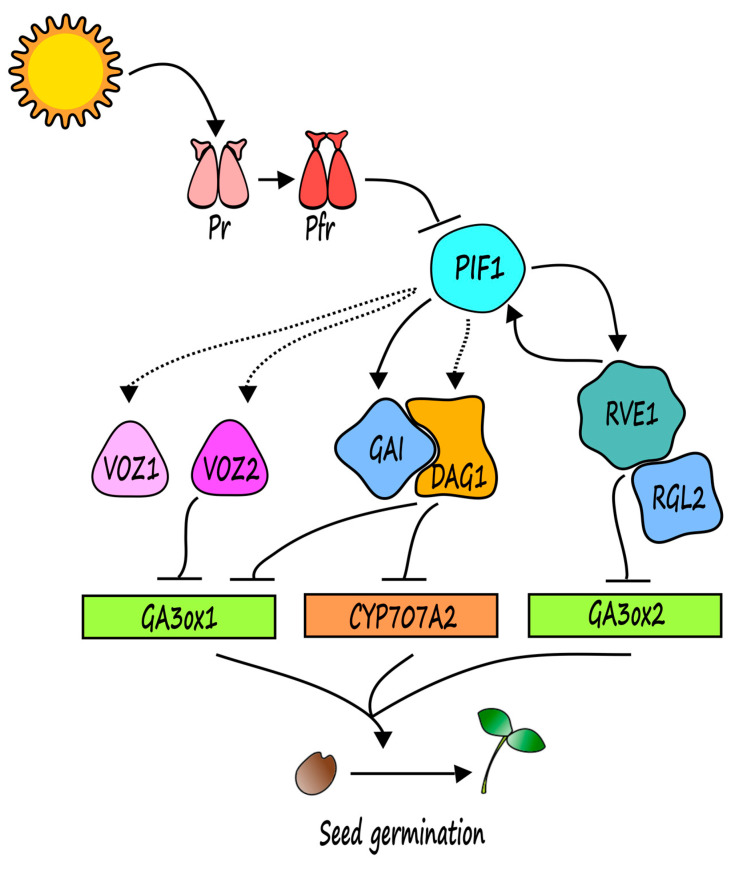
An integrated model of the molecular pathway underlying light-mediated seed germination. Updated model of the phyB-PIF1 signaling pathway regulating the seed germination process. PIF1 directly activates *GAI* and *RVE1* expression, and in turn RVE1 directly induces *PIF1*. RVE1 interacts with the DELLA protein RGL2 to repress *GA3ox2*, while GAI interacts with DAG1 to repress *GA3ox1.* The transcription factors VOZ1 and VOZ2 also repress *GA3ox1*, downstream PIF1. Dotted arrow, indirect control; full arrow, direct control; Pr, phyB inactive form. Pfr, phyB active form.

**Figure 2 genes-12-00052-f002:**
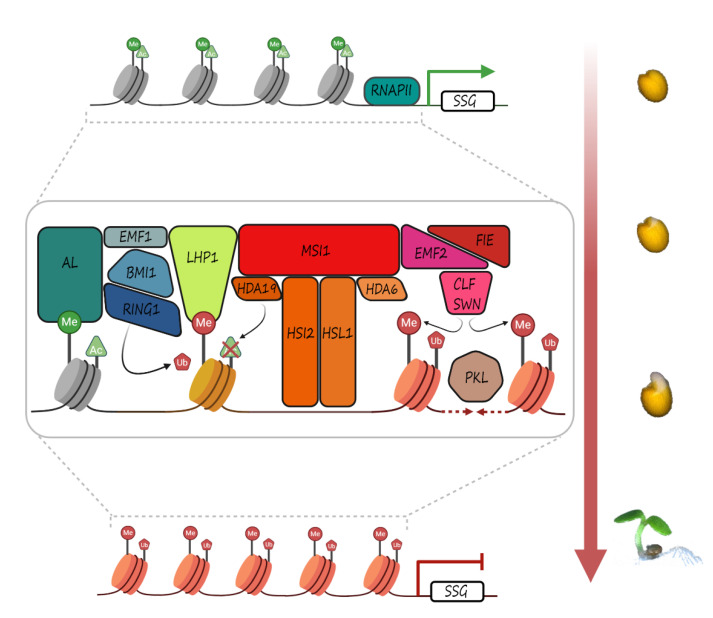
Schematic model of the molecular complexes involved in the control of seed-specific genes. The model depicts the PRC1 and PRC2 complexes and histone deacetylases HDA6 and HDA19 on the promoter of seed-specific genes (SSG). The interacting proteins AL-PHD, HSI2, and HSL1 are also shown. The H3K4me3 activation marker is indicated with green circles, while the H3K27me3 repressing one is indicated with red ones; the acetylation activating marker is shown as a green triangle; H2Aub is reported as a red pentagon. On the right the developmental steps are also reported.

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
