# Peer review of "From the Outside to the Inside: New Insights on the Main Factors That Guide Seed Dormancy and Germination"

_genes, 2020, doi:10.3390/genes12010052_

Round 1

Reviewer 1 Report

In their manuscript, the Authors reviewed recent advances on the complex molecular mechanisms underlying the germination process. They elucidated different regulatory aspects in germination control including light, plant hormones, abiotic stress, epigenetic regulation and translational control, and tried to present an integrated view of this regulatory machine. Undoubtedly, the topic is of interest for the plant biologists’ society. The logic of the manuscript is clear, and it technically sounds. However, my point is that the topic is too broad (the Authors actually consider all together seed development, dormancy, germination and seed-to-seedling transition). Consequently, in the current version, the review does not thoroughly cover all the aspects, therefore, the title and the review itself should be more focused. Otherwise, the MS should be supplemented with more information. My major comments are below.

  1. In the Title, the authors indicate that the review is about the germination process. However, in the abstract they claim, “this review will focus on the different aspects of the molecular control of seed development, dormancy and germination, and also of the seed-to-seedling transition” (lines 20-21). This is misleading. I recommend the Authors to be more specific and to define unequivocally the object they are reviewing.
  2. Despite the abstract claims that “this review will focus on the different aspects of the molecular control of seed development, dormancy and germination, and also of the seed-to-seedling transition” (lines 20-21), there is no comprehensive review of neither seed development nor seed-to-seedling transition in the MS. I encourage the Authors to more specifically focus their review on a certain process (e.g. seed dormancy and germination) or to comprehensively review all processes they claimed.
  3. It is not clear if the title of section 2 means light-phytohormones crosstalk or the role of light and the role of hormonal crosstalk in germination. Anyway, the title does not match the section content since (1) the crosstalk with light is described for GA signaling only, and (2) only the last paragraph (lines 162-175) explicitly describes hormonal crosstalk in post-germinative seedling development (the role of ethylene in ABA/GA balance). The role of ABA in DOG1-mediated dormancy is not specified in the text; therefore, the conclusion “these findings uncovered, at least in part, the molecular pathway which controls seed dormancy, linking ethylene to ABA signaling through ETR1-ERF12/TPL and DOG1” (lines 129-131) sounds obscure. Besides, the Authors do not show that auxin, JAs, CKs, BSs significantly contribute to seed germination as well (e.g. Linkies and Leubner-Metzger, 2012; Miransari and Smith, 2014). Ethylene receptor ETR1 can contribute to light-mediated control of seed germination (Wilson at al., Front Plant Sci, 2014). Etc. Please consider revising section 2. I would suggest splitting it into two sections: one for the hormonal regulation (the internal cues), the other for light-mediated mechanisms (the external signal).
  4. In section 2, the authors talk about light-mediated seed germination but do not specify the role of light in seed germination. It would be more appropriate to at least forward the readers to some of the existing reviews (eg. Yang et al., J Integr Plant Biol, 2020).
  5. In section 3 (lines 177-236), the relation of the text to molecular mechanisms that control seed germination is elusive. Please consider revising.

Besides, I have some minor comments:

  1. Line 67 “The interaction of PIF1 with the active form of phyB induce its phosphorylation”. Please consider rephrasing since in is not clear from this sentence whether PIF1 or phyB is phosphorylated.
  2. Line 82 “recent studies proved that PIF1 interacts with RVE1”. Please indicate the reference.
  3. Line 83 “RVE1 and PIF1 additively control phyB‐83 mediated seed germination”. Please clarify what does additive control mean in this particular case.
  4. Lines 86-87 “a complex signaling pathway with two negative master regulators and a number of downstream elements”. It is not clear for me from neither the text nor the figure what the Authors mean by “two negative master regulators” and “downstream elements”.
  5. Pr and Pfr should be decrypted in the caption for figure 1.

Author Response

  1. In the Title, the authors indicate that the review is about the germination process. However, in the abstract they claim, “this review will focus on the different aspects of the molecular control of seed development, dormancy and germination, and also of the seed-to-seedling transition” (lines 20-21). This is misleading. I recommend the Authors to be more specific and to define unequivocally the object they are reviewing.

We thank the Reviewer for the useful comments. We have revised the manuscript, focusing only on seed dormancy and germination. Accordingly, we have changed the title: “From the outside to the inside: new insights on the main factors that guide seed dormancy and germination”

  1. Despite the abstract claims that “this review will focus on the different aspects of the molecular control of seed development, dormancy and germination, and also of the seed-to-seedling transition” (lines 20-21), there is no comprehensive review of neither seed development nor seed-to-seedling transition in the MS. I encourage the Authors to more specifically focus their review on a certain process (e.g. seed dormancy and germination) or to comprehensively review all processes they claimed.

According to the Reviewer’ suggestion, we focused on seed dormancy and germination. The sentence has been rephrased (lines 20/22).

In addition, we modified the text throughout the manuscript rephrasing sentences only related to seedling development, and highlighting the link to the seed germination process and/or to the underlying molecular processes.

  1. It is not clear if the title of section 2 means light-phytohormones crosstalk or the role of light and the role of hormonal crosstalk in germination. Anyway, the title does not match the section content since (1) the crosstalk with light is described for GA signaling only, and (2) only the last paragraph (lines 162-175) explicitly describes hormonal crosstalk in post-germinative seedling development (the role of ethylene in ABA/GA balance). The role of ABA in DOG1-mediated dormancy is not specified in the text; therefore, the conclusion “these findings uncovered, at least in part, the molecular pathway which controls seed dormancy, linking ethylene to ABA signaling through ETR1-ERF12/TPL and DOG1” (lines 129-131) sounds obscure. Besides, the Authors do not show that auxin, JAs, CKs, BSs significantly contribute to seed germination as well (e.g. Linkies and Leubner-Metzger, 2012; Miransari and Smith, 2014). Ethylene receptor ETR1 can contribute to light-mediated control of seed germination (Wilson at al., Front Plant Sci, 2014). Etc. Please consider revising section 2. I would suggest splitting it into two sections: one for the hormonal regulation (the internal cues), the other for light-mediated mechanisms (the external signal).

We thank the Reviewer for this suggestion. Since the topic of the special issue refers to “new insights”, we mainly presented the most recent evidence. For this reason, the review does not deal with the role of hormones such as auxin, Jas, CKs, BSs, although they play a significant role in the control of seed germination. However, taking into account the Reviewer's comment we added a sentence in section 3, summarizing the involvement of different hormones (lines 131/133). In addition, we added recent data on BRs, from Zhao et al., 2019 (lines 133/138), as well as a comment on ETR1 involvement on seed germination under Far red light (Wilson et al., 2014) (lines 160/165). The role of ABA in DOG1-mediated dormancy has also been specified (lines 149/151).

  1. In section 2, the authors talk about light-mediated seed germination but do not specify the role of light in seed germination. It would be more appropriate to at least forward the readers to some of the existing reviews (eg. Yang et al., J Integr Plant Biol, 2020).

We thank the Reviewer for this suggestion. Since we have mentioned the role of light in seed germination in the introduction, we did not state it in section 2. According to the Reviewer suggestion we have now specified it also in section 2, with the suggested reference (lines 69/71).

  1. In section 3 (lines 177-236), the relation of the text to molecular mechanisms that control seed germination is elusive. Please consider revising.

We agree on this issue raised by the Reviewer; nevertheless, it should be considered that some of the studies we referred to, only present data relative to the seed germination phenotype, overlooking the molecular mechanisms. However, where it was possible, we added the data linking the phenotype to the seed germination molecular pathways (lines 275/278; 303/310).

Besides, I have some minor comments:

  1. Line 67 “The interaction of PIF1 with the active form of phyB induce its phosphorylation”. Please consider rephrasing since in is not clear from this sentence whether PIF1 or phyB is phosphorylated.

The sentence has been modified “The interaction of PIF1 with the active form of phyB induce PIF1 phosphorylation”

  1. Line 82 “recent studies proved that PIF1 interacts with RVE1”. Please indicate the reference.

Done

  1. Line 83 “RVE1 and PIF1 additively control phyB‐83 mediated seed germination”. Please clarify what does additive control mean in this particular case.

According to the Reviewer suggestion, we have added the following sentence: “since pif1rve1 double mutant seeds showed higher germination rates than the single mutants”

  1. Lines 86-87 “a complex signaling pathway with two negative master regulators and a number of downstream elements”. It is not clear for me from neither the text nor the figure what the Authors mean by “two negative master regulators” and “downstream elements”.

We thank the Reviewer for this useful comment; we have modified the sentence as follows: “thus outlining a complex signaling pathway with PIF1 and RVE1 as upstream master regulators which control target genes as GA3ox1 and GA3ox2, through the activity of theDELLA proteins RGL2 and GAI”. In addition, we have modified the legend of figure 1, adding a more detailed description.”

  1. Pr and Pfr should be decrypted in the caption for figure 1.

Done

Reviewer 2 Report

The review is well-written in general. It summarizes different molecular mechanism that regulating seed germination process. The resolution and quality of figures need to be improved. This review does not discuss the potential research directions of this area, which is important for the readers.

Author Response

Comments and Suggestions for Authors

The review is well-written in general. It summarizes different molecular mechanism that regulating seed germination process. The resolution and quality of figures need to be improved. This review does not discuss the potential research directions of this area, which is important for the readers.

We thank the Reviewer for the comments. The resolution of the figures was 600dpi for figure 1, and 300dpi for figure 2, as required by the journal; it is possible that the figures added within the manuscript lost quality. For this reason, we added both figures also as original files.

According to the Reviewer’ suggestion, we added a paragraph in Conclusions dealing with future directions (lines 525/528).

Round 2

Reviewer 1 Report

The authors addressed all the comments. The current version of the review focused, clear and of current interest.